# Rationale of Immunotherapy in Hepatocellular Carcinoma and Its Potential Biomarkers

**DOI:** 10.3390/cancers11121926

**Published:** 2019-12-03

**Authors:** David Tai, Su Pin Choo, Valerie Chew

**Affiliations:** 1National Cancer Centre, Singapore 169610, (NCCS), Singapore; david.tai.w.m@singhealth.com.sg (D.T.); choo.su.pin@singhealth.com.sg (S.P.C.); 2Curie Oncology, Mount Elizabeth Novena Specialist Centre, Singapore 329563, Singapore; 3Translational Immunology Institute (TII), SingHealth-DukeNUS Academic Medical Centre, Singapore 169856, Singapore

**Keywords:** immunotherapy, biomarkers, combination immunotherapy, immune-related adverse events (irAEs), hepatocellular carcinoma (HCC)

## Abstract

Hepatocellular carcinoma (HCC), the most common type of liver cancer, is derived mostly from a background of chronic inflammation. Multiple immunotherapeutic strategies have been evaluated in HCC, with some degree of success, particularly with immune checkpoint blockade (ICB). Despite the initial enthusiasm, treatment benefit is only appreciated in a modest proportion of patients (response rate to single agent ~20%). Therapy-induced immune-related adverse events (irAEs) and economic impact are pertinent considerations with ICB. It is imperative that a deeper understanding of its mechanisms of action either as monotherapy or in combination with other therapeutic agents is needed. We herein discuss the latest developments in the immunotherapeutic approaches for HCC, the potential predictive biomarkers and the rationale for combination therapies. We also outline promising future immunotherapeutic strategies for HCC patients.

## 1. Introduction

Cancer immunotherapy is a rapidly evolving field, which has revolutionized the treatment landscape in oncology this past decade [1]. Unlike conventional cancer therapies, immunotherapeutic approaches do not directly target tumor cells; instead, they target the patient’s immune system or the tumor microenvironment (TME) [2]. A variety of strategies have been explored: cytokine administration, cancer vaccines, adoptive cellular therapy, and immune checkpoint blockade (ICB) [3]. Among which, ICB have been the focus of cancer immunotherapy due to its promising outcomes across multiple advanced solid malignancies, including hepatocellular carcinoma (HCC) [4,5]. HCC is the most common type of primary liver cancer. It is the sixth most common cancer type and the fourth leading cause of cancer death worldwide [6]. Survival after curative surgery remains relatively low. Five-year disease-free survival rates after resection ranges between 24% and 36%, with recurrence rates being as high as 70% [7,8,9]. Before the emergence of immunotherapy, therapeutic development in advanced HCC has been limited partly due to its complex and heterogeneous disease etiologies [10].

The response rate for HCC patients treated with single-agent ICB is modest, at ~15–20%. Moreover, 15–25% of these ICB-treated patients experienced grade 3/4 treatment or immune-related adverse events (TRAEs or irAEs), such as rash, pruritus, diarrhea, and an increase in aspartate aminotransferase (AST) and alanine aminotransferase (ALT) [11,12]. Therefore, a better understanding of mechanistic properties of ICB and predictive biomarkers of response and toxicities is crucial for improved treatment in HCC. This review highlights the current knowledge of immunotherapy in HCC, with particular focus on ICB and the growing understanding of biomarkers discoveries. We also endeavor to provide rationale for combination strategies with ICB and perspectives on personalized immuno-therapeutics for HCC.

## 2. Current Landscape of Immunotherapy in HCC

### 2.1. Immune Checkpoint Blockade (ICB) Therapy

The key mechanism of action for ICB is to block the immune exhaustion or inhibitory pathways induced by chronic immune response against tumor antigen, in order to reactivate the antitumor immune response [13,14]. Immune checkpoint inhibitors are monoclonal antibodies designed to target multiple checkpoint molecules, such as PD-1, CTLA-4, Tim-3, Lag-3, and VISTA, expressed primarily by T cells, as well as PD-L1, the ligand for PD-1, expressed primarily by the tumor or other immune cells [14]. PD-1, PD-L1, and CTLA-4 inhibitors are the most widely evaluated ICB therapies in clinical trials for various solid cancers, including HCC. A summary of the major clinical trials using ICB as monotherapy in HCC, their response rates, and rates of >grade 3 irAEs are provided in Table 1. Combination strategies utilizing ICB are described in greater detail below, in Section 2.4.

#### 2.1.1. Anti-PD-1 Therapy

Two phase I/II clinical studies in HCC, CheckMate040, and Keynote224, using anti-PD-1 monoclonal antibodies nivolumab and pembrolizumab respectively, have been reported [11,12]. CheckMate040, is a multicohort phase I/II, open-label, dose escalation, and expansion trial, using nivolumab alone or in combination with ipilimumab (anti-CTLA4 monoclonal antibody). In cohort 1 and 2 of Checkmate040 (cohort 1: all 214 patients and cohort 2: 85 Asian patients), patients with advanced HCC who were treatment naïve or progressed/intolerant to sorafenib were treated with nivolumab. Keynote224 was a phase II, open-label trial that assessed the efficacy and safety of pembrolizumab in 104 patients with advanced HCC, who were previously treated with sorafenib. Nivolumab demonstrated an objective response rate (ORR) of 20% and disease control rate (DCR) of 64%, whereas pembrolizumab showed an ORR of 17% and DCR of 62% (see Table 1) [11,12]. A subsequent Asian cohort analysis from CheckMate040 demonstrated an ORR of 15% [15]. Both of these trials demonstrated superior ORR and DCR compared to historical responses of sorafenib in advanced HCC (~2% ORR in the SHARP trial) [16]. In addition, the median duration of response was up to 17 months reported in sorafenib experienced HCC patients treated with nivolumab [11], underlining the durability of control in a proportion of patients. Both CheckMate040, and Keynote224 reported moderate (15–25%) >grade 3/4 irAEs (see Table 1). The response rates of anti-PD1 therapy in HCC is, however, modest compared to other cancers, like melanoma (ORR 44%) [17] and renal cell carcinoma (ORR 25%) [18].

Despite encouraging results obtained from initial single-arm studies, two phase III trials in advanced HCC: CheckMate459 (NCT02576509), and Keynote240 (NCT02702401), using nivolumab and pembrolizumab respectively, failed to meet their predetermined primary endpoints of overall survival (OS). In Keynote240 trial, pembrolizumab, when compared to placebo in advanced HCC patients previously treated with sorafenib, did not meet the predetermined dual primary endpoints of improved OS (HR: 0.78; one-sided *p* = 0.0238) and PFS (HR: 0.78; one-sided *p* = 0.0209). Of note, however, the ORR of 18.3% was comparable to earlier studies with median duration of response of 13.8 months [19]. CheckMate459 trial, which compared nivolumab versus sorafenib as first-line treatment in patients with unresectable HCC, also did not meet its prespecified primary endpoint of OS [20]. Median OS was 16.4 months for nivolumab and 14.7 months for sorafenib (HR, 0.85 [95% CI, 0.72–1.02]; *p* = 0.0752). An improvement in ORR was observed with nivolumab compared with sorafenib (odds ratio (95% CI), 2.41 (1.48–3.92)) (see Table 1). Grade 3/4 treatment related adverse events were reported in 81 patients (22%) in the nivolumab arm and 179 patients (49%) in the sorafenib arm [20]. Despite both studies not meeting their primary endpoints, there was a clear trend toward improved OS in favor of ICB. Nevertheless, treatment effect of single-agent ICB appears binary with a modest proportion of patients truly deriving benefit. This underlines the need for a predictive biomarker of response as well as rational combination strategies.

#### 2.1.2. Anti-PD-L1 Therapy

Several anti-PD-L1 monoclonal antibodies are currently under clinical trials in advanced HCC include avelumab, durvalumab, and atezolizumab. Avelumab monotherapy is currently being evaluated in a phase II study (NCT03389126). Durvalumab monotherapy was evaluated in a phase I/II trial in various solid tumors and reported an ORR of 10.3% in 39 HCC patients who declined, were intolerant, or progressed on prior sorafenib [21] (see Table 1). Atezolizumab monotherapy was compared against combination of atezolizumab and bevacizumab (anti-VEGF antibody) in advanced HCC patients in the Arm F of Phase Ib GO30140 study [22]. Median progression-free survival (PFS) was 3.4 months in the monotherapy arm, compared to 5.6 months in the combination arm (HR 0.55, *p* = 0.018) [22].

#### 2.1.3. Anti-CTLA-4 Monoclonal Antibodies

Anti-CTLA4 antibody (Ipilimumab) was first approved by FDA in 2011 for the treatment of melanoma, following the result from the phase III trial, showing significant overall survival benefit compared to gp100 vaccine alone [23]. Another anti-CTLA4 antibody, tremelimumab, was evaluated for safety, antitumor, and antiviral activity in HCV-related HCC as monotherapy in a single-arm phase II trial (NCT01008358) [24]. An ORR of 17.6% was reported among 17 patients (see Table 1) as well as anti-HCV viral immunity [24]. Result from another phase I/II study of durvalumab and tremelimumab in patients with unresectable HCC (NCT02519348) will be announced in the near future [25].

### 2.2. Current Knowledge on Biomarkers for ICB and Its Relevance in HCC

Predictive biomarkers of response in ICB across different cancer types have been extensively reviewed [26,27,28]. We summarize the key biomarkers from intratumoral tissues (tumor or TME specific tissue markers) and extratumoral tissues (from peripheral blood, serum or feces) in Table 2 and provide evidence and perspectives, where available, on HCC.

#### 2.2.1. PD-L1 Expression

PD-L1 expression is one of the earliest and most widely used biomarkers of response to immunotherapy. PD-L1 IHC is approved by FDA as a companion diagnostic when considering use of anti-PD1 therapy in NSCLC [29,30]. Despite this, the utility of PD-L1 expression across multiple tumor types has been disparate: some with positive association [31,32,33,34,35], while others with no association [11,12,18,36,37] with clinical outcome. Within HCC tissues, PD-L1 was found to be expressed by the tumor cells [38] and macrophages [39], both of which were associated with poor post-resection prognosis; meanwhile, PD-1 was expressed mainly by the T cells, including regulatory T cells (Treg) [40,41]. It has also been shown that the PD-L1 expression in HCC is generally low (~10% by tumor cells) and highly heterogeneous across different anti-PD-L1 staining antibodies used [42]. Indeed, tumor PD-L1 expression was not a robust biomarker for response to anti-PD-1 therapy in both CheckMate040 and Keynote224 trials in HCC [11,12]. Reasons for contradictory results from clinical trials using PD-L1 as a biomarker include the different assays for detection, the spatial heterogeneity in expression of PD-L1, and various standards and cutoffs used in assessing positive staining [29,30,43].

Another important consideration is that nontumor host cells could also express PD-L1 and be considered as the biomarker for response to anti-PD-1/anti-PD-L1 ICB [44]. For instance, studies in melanoma [45], urothelial carcinoma [46], and HCC [12] have found that PD-L1 expression on nontumor host cells, such as TILs, was associated with response to anti-PD1 or anti-PD-L1 therapy. More recently, circulating exosomal PD-L1 was shown to correlate with clinical response to anti-PD-1 therapy, in a study conducted in patients with advanced melanoma [47]. Increased circulating exosomal PD-L1 was indicative of adaptive response by the tumor cells to T-cell reinvigoration [47]. One recent preclinical study in mice models demonstrated that, by suppressing exosomal PD-L1, antitumor immune response and memory could be induced even in the anti-PD-L1-resistant models [48]. Given the high intratumoral heterogeneity of HCC tumors as described previously [49,50], exosomal markers could serve as an attractive biomarker to predict clinical outcome to immunotherapy.

Recent research focuses also on the post-translational regulation of PD-L1 expression [51,52,53]. For instance, epigenetic regulation of PD-L1 protein expression by microRNA has been implicated in various cancers [51]. Maintenance of PD-L1 on the cell membrane and prevention from its lysosomal degradation by regulatory proteins, such as CMTM6, could also play an important role [54]. Additionally, given the link between inflammation and IFNγ-induced upregulation of PD-L1 expression in tumor [55,56], IFNγ signature has also been shown to be a biomarker of response to ICB in multiple cancer types [57,58]. Such data in HCC is currently lacking.

#### 2.2.2. Tumor Mutational Burden (TMB) and Specific Genomic Mutations

Tumor mutational burden (TMB) correlates with responses with ICB across multiple cancer types, including HCC [59,60,61,62,63]. One cross cancer study on TMB indicated that tumors with high TMB would also have higher expression PD-L1 on tumor cells, predicting their response to anti-PD-1/PD-L1 therapy [63]. Indeed, tumors with high TMB are associated with more neoantigens and linked to a more inflamed tumor microenvironment, higher IFNγexpression, and upregulation of PD-L1 expression [55,108]. In this study, TMB level is considered moderate for HCC, consistent with a modest response rate to ICB in HCC [63]. However, a separate small case series of 17 HCC patients treated with anti-PD-1 ICB showed no significant association between TMB and response [64]. Furthermore, DNA mismatch repair (MMR) gene deficiency, which results from a heavy mutational burden and predictive of response to immunotherapy, is infrequent in HCC [109].

Specific tumor mutations, such as Wnt alteration/β-catenin mutation, are linked to a T-cell exclusion or immunosuppressive TME and resistance to ICB in patients with advanced melanoma [66,67]. A study involving 27 HCC patients who received ICB (a mix of anti-PD1/anti-PD-L1/anti-CTLA4 or combination) found Wnt-pathway mutation to be predictive of resistance to therapy [68]. A more recent single-cell RNA sequencing study on biopsy samples taken from 19 liver cancer patients (9 HCC and 10 cholangiocarcinoma patients) treated with mixed ICB regimens showed that patients with less transcriptomically diverse tumors demonstrated a better response and survival profile [65]. This study also identified VEGFA as one of the possible mechanisms of resistance to ICB, hence providing rationale for combination of ICB with an anti-angiogenic agent [65]. However, it is not known how TMB is related to the transcriptomic diversity of tumors. Further studies are needed to clarify the relevance of TMB and specific molecular alterations (e.g., Wnt alteration/β-catenin mutation or VEGF overexpression) in relation to response to ICB.

#### 2.2.3. Tumor-Infiltrating Lymphocytes (TILs) Density and Phenotypes

Density of TILs, particularly CD8+ T cells, connotes a better prognosis in various cancer types, including HCC [110,111,112,113,114]. Several studies have shown that higher TILs density, particularly for CD8+ T cells, predicts for better survival after ICB [45,69,70,71]. CD8+ T-cells density at the invasive margin, and not at the center of the tumor, was the most important determinant of better outcomes in melanoma patients treated with anti-PD1 ICB [69]. In addition, T-cell receptor (TCR) diversity or clonality, indicative of its ability to recognize diverse repertoire of tumor antigens, has also been shown to correlate with response to ICB [69,73,74,75]. As recognition of tumor antigens depends on antigen-presentation components, it is hence not surprising that HLA diversity predicts better responses to immunotherapy [76].

Apart from density and location of TILs, their phenotypes also play an important role. For instance, the cytolytic property of T cells, indicated by expression of pro-inflammatory genes perforin and granzyme, was associated with response to anti-PD-1 therapy in melanoma patients, despite no significant change in TILs density [73,77]. A study using single-cell RNA sequencing (scRNA seq) technologies to profile TILs found that the ratio of activated to exhausted CD8+ T cells in the tumor correlated with the response to ICB in melanoma patients [78]. The recent scRNA seq study in liver cancer patients treated with mixed ICBs also concurred with these findings that tumor-infiltrating cytolytic T cells play an important role in predicting response to immunotherapy [65]. Another immunoprofiling study in HCC cohort who received preoperative ICB treatment, followed by resection, showed that an increase in effector T cell was associated with complete response [72]. Both studies underlined the importance of TILs, particularly its phenotypes, as predictive biomarker of response in HCC patients. In fact, it was previously shown that only ~20% of HCC tumors were considered well infiltrated by immune cells [67,115], consistent with the reported clinical outcomes in anti-PD-1 ICB monotherapy trials.

Other immune subsets such as regulatory T cells (Treg) or macrophages also have predictive values for response to ICB. For instance, higher frequency of Treg, myeloid derived suppressor cells (MDSCs) [79] and tumor-associated macrophages (TAM) [81] are linked to unresponsiveness to ICB. Treg has been linked to cancer hyper-progression after ICB [80], further underlining its important regulatory role in ICB response. Their roles in HCC remain to be elucidated.

#### 2.2.4. Peripheral Immune Cells’ Phenotypes

Peripheral blood is an important biological material for monitoring clinical response after ICB. As T cells are the primary targets for ICB, the pretreatment diversity of TCR repertoire is an important biomarker of response to ICB in the circulating blood [82,83,84]. The phenotypes of T cells have also been studied, and the ratio of reinvigorated CD8+ T cell to the tumor burden [85], as well as the activation status of both CD4+ and CD8+ T cells [86], upon treatment could predict for response to ICB in melanoma patients. Other circulating immune cells, such as immunosuppressive MDSCs, have been shown to correlate with poor response to anti-CTLA-4 therapy in multiple studies in melanoma patients [87,88,89,90]. The ratio of neutrophil to lymphocytes was associated with decreased PFS and OS after ICB treatment [91,92,93,94].

The role of circulating Treg cells is, however, controversial. Higher baseline frequency of Treg has been linked to disease control after ICB [84]. In two studies in patients with advanced melanoma treated with ipilimumab, one reported that an on-therapy increase in frequency of circulating Treg at week 6 was associated with improved PFS [95]. In contrast, another study reported that a decrease in frequency of circulating Treg at a later timepoint of week 12 was associated with disease control upon ICB [96]. It is possible to speculate that an initial increase followed by decrease in Treg might be a sign of clinical response to immunotherapy.

It is therefore important to study the dynamic changes of various peripheral immune subsets at defined time points after immunotherapy, for a more accurate comparison. Such studies are currently lacking in HCC.

#### 2.2.5. Other Extratumoral Biomarkers

Other noncellular biomarkers in the blood include lactate dehydrogenase (LDH), an enzyme that is released by rapidly growing tumors and associated with large tumor burden, tumor hypoxia, angiogenesis, and worse prognosis [116,117]. High baseline serum LDH levels are associated with worse outcomes with ICB [89,97,98,100]. Dynamic changes of LDH levels while on treatment could also predict outcomes. On-treatment reduction in LDH levels was associated with better response in patients with advanced melanoma treated with ipilimumab [96,99]. As serum LDH level has been used as a biomarker in predicting response to TACE [118] and sorafenib [119] in HCC, its role in predicting responses to ICB would be of interest.

As described earlier, increased circulating exosomal PD-L1 during early stages of anti-PD-1 therapy positively correlated with clinical response in melanoma patients [47]. Apart from this, circulating cell-free DNA (cfDNA) carrying tumor-related genetic and epigenetic alterations have been shown to be related to cancer development, progression, and resistance to therapy [120]. This makes cfDNA an easily accessible biomarker to predict tumor response to therapy, which is potentially not affected by intratumoral heterogeneity [121,122]. In fact, specific mutations or TMB can be detected from circulating cfDNA and that have been shown to associate with responses to ICB [101,102]. One particular study found that hypermutated circulating tumor DNA was associated with clinical outcome in 69 cancer patients, including three HCC patients, treated with a variety of ICBs [101].

Lastly, the gut microbiome analyzed from the feces also seem to play an important role in determining response to ICB. In fact, the role of microbiota in human health and disease, particularly in cancer, has been increasingly appreciated [123,124]. Interestingly, different strains of microbiome have been found to be associated with response to ICB in four major reports on baseline fecal sample analysis from melanoma [104,105,106] and other cancer types [107]. Of note, transferring the response-associated gut microbiota to germ-free or antibiotic-treated mice could induce ICB response, making fecal transfer an area of intense research interest at present. Other than the specific microbiota strain, the general increase in microbiota diversity [106] and the ratio of response-associated to resistance-associated microbiome [103] were also associated with better response to ICB. It remains to be determined if such a microbiome is related to response to ICB in HCC patients.

### 2.3. IrAEs and Its Association with Outcomes of ICB in HCC

Data from several key clinical trials using ICB in HCC patients showed that 15%–45% of the patients may experience grade 3 or greater treatment-related AEs, most of which being irAEs (see Table 1). Association between incidence of irAEs and clinical outcomes with ICB are conflicting. Overall irAEs have been found to be associated with better clinical outcomes in both melanoma and NSCLC patients treated with nivolumab [125,126]. Some studies, however, reported no association between irAEs and clinical outcome in selected malignancies [127,128]. Interestingly, a retrospective study of patients with various nonmelanoma cancers who received anti-PD-1 therapy demonstrated that only low-grade irAEs were associated with better responses in these patients [127]. Some studies even suggested that cancer-specific irAEs may be important in determining response to immunotherapy. The association of vitiligo to better responses in melanoma patients [129] and thyroid toxicity with better outcomes in NSCLC patients [130] with ICB are two such examples. A recent study on 114 HCC patients treated with mixed ICB reported a correlation of irAEs with higher DCR, median PFS and OS [131]. A future study involving a larger number of HCC patients with better defined immunotherapy regimens would be necessary to have a more conclusive assessment.

Several studies, with the majority of them in ipilimumab-treated melanoma patients, have reported various predictive biomarkers for irAEs, such as level of circulating IL-6, autoantibodies, blood-cell counts, T-cell repertoire, and gut microbiome [132]. For instance, the level of baseline circulating IL6 and being female are associated with higher incidences of irAEs in ipilimumab-treated advanced-melanoma patients [133]. A retrospective review of 167 patients with various solid tumor types treated with nivolumab or pembrolizumab suggested that patients with higher baseline lymphocyte counts have a greater risk for irAEs [134]. Another study on a group of 101 Japanese melanoma patients treated with nivolumab showed that the increase in total white-blood-cell count and decrease in relative lymphocyte count at the point of or just prior to irAEs were associated with lung and gastrointestinal irAEs [135]. Putting these two studies together, a higher baseline levels of lymphocytes predispose the patients to irAEs and the decrease of lymphocytes prior to or during the event of irAEs could indicate relocation or recruitment to the site of toxicities. The pretreatment or on-therapy level of autoantibodies, which is a known factor for autoimmune diseases, has also been implicated as a predictive biomarker for the development of irAEs in various cancer types with ICB [136,137,138,139]. There remains no study evaluating the association of irAE response with ICB in HCC. It would be interesting to note whether liver-specific toxicities would be related to response to immunotherapy in HCC.

### 2.4. Current Landscape and Rationale of Combination Immunotherapy in HCC

A combination of four major factors are needed to achieve an effective and sustained immune response: (1) release of tumor-specific antigens to induce T-cell response; (2) adequate generation of tumor-specific cytotoxic T cells with effective trafficking into TME; (3) appropriate TME remodelling strategies; and (4) overcoming exhaustion pathways which inevitably follows after the local immune activation (Figure 1). We next provide the rationale for various combination therapies currently pursued in HCC by ascribing to these factors. A list of current combination therapy involving ICB in HCC are listed in Table 3.

#### 2.4.1. ICB and ICB Combination

It is known that anti-PD-1 and anti-CTLA-4 antibodies have differences in their underlying functional mechanisms [13,148]. For instance, anti-PD-1 ICB was thought to act primarily at the interface of T cells and tumor cells within the local tumor microenvironment, while anti-CTL4 ICB was shown to be able to act more upstream at the phase of T cells priming at the lymph nodes [13,148]. Hence, this combination was based on its potential synergistic antitumor activity [149]. A combination of nivolumab and Ipilimumab, which was first evaluated in a phase III trial for patients with advanced melanoma, demonstrated superior outcomes in terms of both progression-free survival and median survival compared to monotherapy with nivolumab or ipilimumab alone [17]. This provided impetus for other solid tumors, including HCC. The third arm of CheckMate 040 evaluated combination nivolumab and ipilimumab in 148 sorafenib-treated patients. Subjects were randomized to three arms: [A] NIVO 1 mg/kg + IPI 3 mg/kg Q3W (4 doses) or [B] NIVO 3 mg/kg + IPI 1 mg/kg Q3W (four doses), each followed by NIVO 240 mg Q2W, or [C] NIVO 3 mg/kg Q2W + IPI 1 mg/kg Q6W. The overall response rate was 31%, with seven complete responses (see Table 3). The 24-month OS rate was 40%, with 37% of patients having a grade 3–4 irAEs (most common all-grade adverse events were pruritus and rash) [140] and 5% having grade 3–4 adverse events, leading to discontinuation. Encouraging results prompted the commencement of CA209-9DW, a phase 3 trial comparing combination ipilimumab and nivolumab against sorafenib or lenvatinib in treatment naïve advanced HCC.

Combination of durvalumab (anti-PD-L1 antibody) and tremelimumab (anti-CTLA4 antibody) (NCT02519348) is currently being evaluated in a Phase I/II study. Preliminary results based only on 40 patients showed a modest ORR of 15% [25] (see Table 3). A large multicenter phase III trial of durvalumab and tremelimumab as first-line treatment in patients with unresectable HCC: HIMALAYA study (NCT03298451) [141] with estimated enrolment of 1310 patients is currently ongoing.

#### 2.4.2. ICB and Anti-Angiogenesis Agent

Angiogenesis, one of the hallmarks of cancer, leads to leaky vasculature, hypoxia, and activation of multiple immunosuppressive pathways in TME as a consequence of rapid tumor growth [150,151,152]. An anti-angiogenic agent aims to normalize the intratumoral vasculature, hence restoring the equilibrium toward a less protumoral or less immunosuppressive TME [153,154]. The role of vascular endothelial growth factor (VEGF) in driving tumor angiogenesis has made it an attractive therapeutic target. Bevacizumab, a humanized monoclonal antibody against VEGF, has gained FDA approval for many advanced malignancies [155]. The multiple roles of VEGF in reprogramming the tumor microenvironment have been discussed in depth previously [154]. Chiefly, VEGF plays an important role in immunosuppressive regulatory T cells’ (Treg) recruitment into the tumor. VEGF inhibition is purported to enhance local antitumor immunity by reducing accumulation of Treg [156]. It was also previously shown that anti-angiogenic agents can increase infiltration of adoptively transferred T cells into a tumor [157]. In a recent study using murine models of HCC, it was shown that this combination therapy reprogrammed the TME by increasing cytotoxic CD8 T cell, while reducing Treg infiltration in HCC tissue and shifting the M1/M2 macrophages ratio in favor of antitumoral TME [158]. A randomized study evaluating atezolizumab (anti-PD-L1 therapy) as monotherapy vs. the combination of atezolizumab + bevacizumab (anti-VEGF therapy; Arm F), as well as single-arm atezolizumab + bevacizumab (Arm A) from a Phase 1b GO30140 study, was conducted in advanced HCC patients and suggested superiority of combination therapy [22]. Concurrently, the outcome from IMbrave150 (NCT03434379) a Phase III, open-label, multicenter, randomized study evaluating combination atezolizumab and bevacizumab versus sorafenib in patients with locally advanced or metastatic and/or unresectable HCC was recently announced [142]. This study met its co-primary endpoints of demonstrating statistically significant and clinically meaningful improvements in both PFS and OS in favor of combination atezolizumab and bevacizumab [142]. With increasing appreciation of immune-modulatory properties of targeted therapies, future combinations of immunotherapy and targeted therapy based on strong rationale and well-studied mechanism of actions would be paramount for drug development.

#### 2.4.3. ICB and Multitargeted Tyrosine Kinase Inhibitors (mTKIs)

Sorafenib, an oral multitargeted tyrosine kinase inhibitor (mTKIs), has been the only systemic therapy for treatment of advanced HCC following the successful SHARP trial in 2008 [16]. Targets of Sorafenib include VEGFR, PDGFR, and RAF kinases, hence exerting antitumor effects through anti-angiogenesis, antiproliferation, and pro-apoptosis [159]. The impact of mTKIs on the TME has also been discussed before [160,161,162]. Most studies demonstrated the immunomodulating properties of mTKIs, such as reduction of MDSCs and Treg [163,164,165,166], enhancing T and NK cells tumor infiltration and activation [167,168], and boosting antitumor immune response. Studies have also discussed the immuno-modulatory properties of mTKIs which could synergize with immunotherapy [169,170]. Furthermore, tumor-cell death induced by mTKIs could serve as a source of tumor antigens that could then activate the specific T cells capable of more cell killing (see Figure 1). Besides that, angiogenesis is one of the common targets for these mTKIs, as well. Two large randomized studies in front-line systemic therapy employs this strategy. Combination atezolizumab (anti-PD-L1 therapy) + cabozantinib (mTKIs) in the COSMIC-312 trial (NCT03755791) [143] and combination of pembrolizumab (anti-PD-1 therapy) and lenvatinib (mTKIs) in the LEAP-002 trial (NCT03713593 or Keynote 524) are currently enrolling [144,145]. Twenty-two systemic treatment-naïve HCC patients were treated with combination avelumab (anti-PD-L1 therapy) and axitinib (mTKIs) with an ORR of 13.6% and median progression-free survival (mPFS) of 5.5 months (see Table 3) [146]. However, toxicities of this combination might be a concern. Grade 3/4 treatment-related adverse events were reported to be 72.7%. Eleven (50%) patients encountered grade 3/4 hypertension, and 22.7% experienced grade 3/4 palmar–plantar erythrodysesthesia (PPE) [146].

#### 2.4.4. Other ICB Combinations

Release of tumor antigen upon tumor-cell killing by chemotherapy, radiotherapy, or transarterial-chemoembolization (TACE) [171,172] further enhances immunogenic cell death. This provides the rationale for combination strategies with ICB (Figure 1). Potential immunogenic cell death induced by oxaliplatin-based chemotherapy containing FOLFOX4 (infusional fluorouracil, leucovorin and oxaliplatin) or GEMOX (gemcitabine and oxaliplatin) provides rationale for an ongoing phase II study in combination with camrelizumab (an anti-PD-1 antibody) in advanced HCC and biliary tract cancer [147]. A number of clinical studies evaluating combination radiotherapy with ICB are in progress [173]. One study in HCC patients treated with external beam RT (EBRT) showed an increase in soluble PD-L1 level post-treatment [174]. Another study, using selective internal radiotherapy (SIRT) in HCC patients, reported enhanced immune cell activation and recruitment, particularly ones that express checkpoint molecule PD-1 [175]. Both studies suggest that combination radiotherapy with ICB could be synergistic. Other locoregional therapies, like transarterial chemoembolization (TACE), have also been explored in combination with immunotherapy. For instance, a multicenter pilot study evaluating the safety of combination of nivolumab with drug-eluting bead-TACE (deb-TACE) in patients with HCC is currently underway [176]. Another study evaluating the safety and efficacy of combination treatment with pembrolizumab and TACE is also ongoing (NCT03397654). Apart from anti-PD-1 therapy, the combination of tremelimumab (anti-CTLA4 therapy) with local therapy (RFA or TACE) has been explored in 32 HCC patients [177].

### 2.5. Other Immunotherapies and Their Potential as Combination in HCC

#### 2.5.1. Adoptive Cell Therapy (ACT)

T cells engineered to express chimeric antigen receptors (CARs), or autologous T cells expanded and engineered ex vivo with specific targeted tumor antigen(s), have been explored as an immunotherapeutic strategy in cancers, including HCC [178,179]. CAR-T cells directed against GPC-3, CEA, or Mucin 1 are currently being evaluated in early phase trials in various solid tumors, including HCC [180]. Of note, T-cell therapy targeting HCC-specific antigens, such as AFP, has been evaluated previously with disappointing outcomes (NCT03349255). Possible explanation behind this lack of activity could be attributable to low T-cell affinities and high expression of PD-1 T-cell exhaustion markers [181].

Other ACTs such as the use of IL-2-activated and -expanded autologous TILs in vitro have demonstrated improved recurrence-free survival (RFS) after resection in 150 HCC patients [182]. In addition, Cytokine-induced killer (CIK) cells, a heterogeneous cytotoxic immune populations consisting of CD8+ T cells, CD56+ NK cells, and CD3+CD56+ NKT cells, was demonstrated to be safe, with a lower recurrence rate and improved RFS and OS in HCC [183]. NK cell therapy has also been explored for HCC treatment, based on findings that NK cells are dysfunctional in HCC and tumor-infiltration with activated NK cells is associated with superior survival in HCC patients [184,185]. More recently, engineered NK cells or CAR-NK cell therapy with tumor specificity are being explored for various cancer types, including HCC [186].

ACT could enhance the frequency of tumor-specific T cells, however, these tumor antigen-specific T cells would migrate to TME and eventually became exhausted given the immunosuppressive state. Therefore, combination with checkpoint inhibitors could potentially reinvigorate the activity of these T cells (Figure 1). Combination ACT with checkpoint inhibitor is yet to be explored in HCC.

#### 2.5.2. Cancer Vaccines

Cancer vaccines either in the form of peptide, dendritic cell-pulsed with synthetic peptide or RNA vectors based on personalized neoantigens have demonstrated promising outcome in patients with advanced melanoma [187,188]. In contrast to that, cancer vaccines targeting individual tumor-associated antigens (TAAs), such as NY-ESO1, glypican-3 (GPC3), and alpha-fetoprotein (AFP), have met with limited success in HCC [189]. This is most likely due to significant intra and inter-tumor genomic heterogeneity, compounded by a highly immunosuppressive TME. For instance, the AFP vaccine showed limited clinical benefit despite detectable T-cell responses [190,191].

To circumvent this, an ongoing trial evaluating therapeutic cancer vaccine IMA970A, a multi-peptide-based HCC vaccine composed of 16 newly discovered and overexpressed tumor-associated peptides (TUMAPs) identified from resected HCC tissues (clinical trial: NCT03203005) was envisioned. It remains to be determined if such multi-peptide cancer vaccines in HCC will be successful. Given the immunosuppressive internal milieu of HCC, it is likely that combinations with other immunotherapeutic agents will be needed (Figure 1). One Phase Ib/II trial using DSP-7888, a novel WT1 Peptide-Based Vaccine, in combination with nivolumab or pembrolizumab for patients with advanced solid tumors including HCC (NCT03311334), is currently enrolling patients.

#### 2.5.3. Oncolytic Virus Therapy

Oncolytic virus therapy involves the use of native or genetically modified viruses that show selective infection, replication and killing of tumors cells [192,193]. These viruses can also be engineered to express immune-stimulatory genes such as GM-CSF, a cytokine which could enhance antitumor immunity by stimulating antigen-presenting cells and promote the tumour infiltration and maturation of NK cells and T cells [194]. Oncolytic virus therapies have been tested in preclinical and phase I/II clinical trials for HCC [195]. For instance, JX-594, an engineered vaccinia virus with thymidine kinase-deactivated, was well tolerated [196] and demonstrated promising outcome in phase II clinical trial in HCC patients [197]. However, a randomized Phase III trial comparing JX-594 versus sorafenib in patients with advanced HCC (PHOCUS) (NCT02562755) halted enrolment recently due to futility. We believe part of the reason for such failures could be due to the immunosuppressive TME of HCC [40]. It is therefore likely that the success of oncolytic virus could be enhanced in combination with ICB (Figure 1). Indeed, several clinical trials using combination of oncolytic virus and ICB are ongoing, including in advanced HCC [198].

## 3. Future Perspectives

Challenges remain in identifying HCC patients who could best benefit from immunotherapy. Based on the biomarker studies in other tumor types (see Table 2), the presence of tumor infiltrating T cells, particularly cytotoxic CD8 T cells, predicts for response to immunotherapy. As HCC tumors are enriched with Treg [40] and generally not well infiltrated by immune cells [67,115], strategies to inhibit Treg and enhance T cells infiltration, in combination with ICB, is important. Given the recent success of Phase III trial in HCC, using ICB plus anti-angiogenesis agent (IMbrave150) [142], it is increasingly clear that a combination strategy with clear scientific rationale is necessary. We also need robust biomarkers from longitudinal tumor and blood sampling, as well as multi-omics interrogation to uncover the intrinsic and acquired resistance mechanisms or incidence of irAEs to these treatments. While we acknowledge the potential of combination immunotherapeutic strategies in future, potential enhanced toxicities, given the coexisting liver dysfunction in HCC patients, are also the main concerns to be considered. Further characterization of irAEs in tandem with various combination strategies is of current utmost importance when treating patients.

## 4. Concluding Remarks

Clinical trials evaluating the use of monotherapy or combination immunotherapeutic agents in HCC are underway. Intensive studies on the mechanisms of actions for evidence-based combination strategies, as well as identification of predictive biomarkers of response and irAEs, are also ongoing. This will result in safer, more effective, and, perhaps, more personalized immunotherapeutic strategies for patients with HCC in the near future.

## Figures and Tables

**Figure 1 cancers-11-01926-f001:**
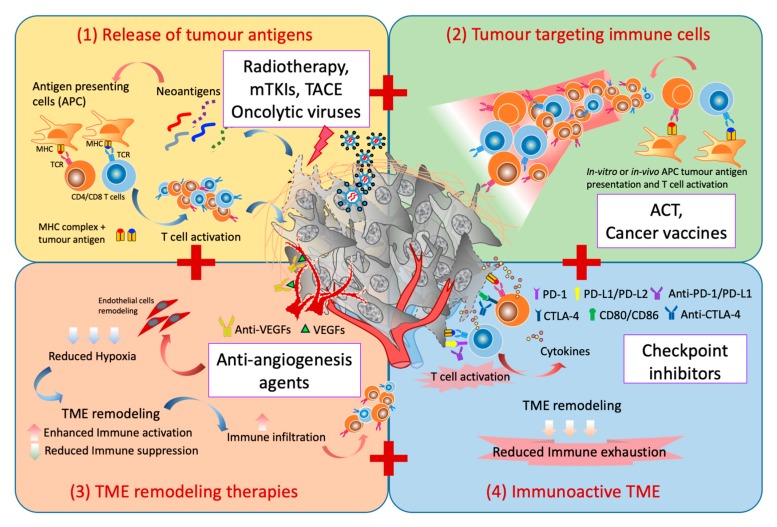
Combination strategies for immunotherapy in HCC. There are four key elements for successful immunotherapeutic strategies: (**1**) the release of tumor antigen to prime the tumor-antigen-specific T-cell response, i.e., the use of radiotherapy, multitargeted tyrokine kinase inhibitors (mTKIs), TACE, or oncolytic viruses that can induce immunogenic cell death; (**2**) the increase in the frequency of tumor-specific cytotoxic T cells which could home into the TME, i.e., by adoptive cell therapy (ACT) or cancer vaccines; (**3**) the TME remodelling strategies such as normalization of the blood to reduce the hypoxic and immunosuppressive microenvironment, i.e., with anti-antiangiogenesis agents; and (**4**) the blocking of the exhaustion pathways which inevitably follows after the local immune activation to reinvigorate the antitumor immune response, i.e., checkpoint-inhibitors.

**Table 1 cancers-11-01926-t001:** Immune checkpoint monotherapy clinical trials in HCC.

Study Name	Phase	Target	Treatments	Estimated Enrolment ^ (*n*)	ORR (%)	DCR (%)	PFS (Median, mo)	OS (Median, mo)	Adverse Effect † > Grade 3 (%)
NCT01658878 (CheckMate040) [11]	I/II	PD-1	Nivolumab	214	20%	64%	4	15.1	25%
NCT01658878 (CheckMate040-Asian cohort analysis)) [15]	I/II	PD-1	Nivolumab	85	15%	49%	NA	14.9	16%
NCT02702414 (Keynote224) [12]	II	PD-1	Pembrolizumab	104	17%	62%	4.9	12.9	15%
NCT02576509 (CheckMate459) [20]	III	PD-1	Nivolumab vs. Sorafenib	743 (371 vs. 372)	15% vs. 7%	55% vs. 58%	3.7 vs. 3.8	16.4 vs. 14.7	22% vs. 49%
NCT02702401 (Keynote240) [19]	III	PD-1	Pembrolizumab vs. placebo	413 (278 vs. 135)	16.9% vs. 4.4%	62.2% vs. 53.3%	3.0 vs. 2.8	13.9 vs. 10.6	18.6% vs. 7.5%
NCT01693562 [21]	I/II	PD-L1	Durvalumab	39	10.3%	32.5% #	2.7	13.2	20%
NCT01008358 [24]	II	CTLA-4	Tremelimumab	20	17.6%	76.4%	6.48 (TTP)	8.2	45%

^, most updated from clinicaltrials.gov as of August 2019; *n*, number of patients; ORR, overall response rate; DCR, disease control rate; PFS, progression-free survival; OS, overall survival; mo, months; †, treatment-related adverse effects; NA, not available; #, CR+ PR + SD > 24 weeks; TTP, time to progression.

**Table 2 cancers-11-01926-t002:** Biomarkers predictive of response of immune checkpoint therapy.

Source	Biomarker	Assay Type	Cancer Type	Clinical Relevance	Relevance to HCC
Intra-tumoral	PD-L1 expression	Immunohistochemistry (IHC)	Multiple	Expression on tumor cells [31,32,33,34,35] or immune cells [12,45,46] showed positive association with response. No significant association with response [11,12,18,36,37].	No association with response [11,12]. Marginal association between PD-L1 expression on nontumor host cell (Keynote224) with response [12].
	IFNγ signature	NGS or targeted genes seq	Multiple	Predictive of response to ICB [57,58].	No direct evidence in HCC yet.CC yet.
	TMB	NGS, WES, or WGS	Multiple	Higher TMB was positively associated with improved response to ICB [59,60,61,62].	Positive association with response (mixed cancer types including HCC) [59,63]. No significant association with anti-PD-1 ICB (17 HCC patients) [64].
	Tumor transcriptomic diversity	Single-cell RNA seq	HCC/iCCA	Lower tumor transcriptomic diversity was associated with PFS and OS of liver cancer patients treated with mixed ICB [65].	
	Wnt/β-catenin pathway mutation	NGS, WES, or targeted genes seq	Melanoma/HCC	Wnt/β-catenin mutation was linked to T cell exclusion, immunosuppressive TME and resistance to ICB [66,67].	Wnt pathway mutation was related to resistance to therapy (27 HCC patients) [68].
	TILs density (hot/cold)	IHC or RNA seq	Multiple	Higher TILs density (particularly CD8+ TILs) was associated with superior clinical response [45,69,70,71].	An increase in CD8+ T-cell tumor infiltration and effector T-cells [72] or cytolytic T cell infiltrates [65] was associated with response to ICB in HCC.
	T-cell repertoire	RNA seq or TCR seq	Melanoma and lung cancer	TIL clonality positively correlated with response [69,73,74,75].	
	HLA diversity	NGS, WES	Melanoma and NSCLC	HLA-I heterozygosity was associated with improved OS after ICB [76].	
	Specific CD8+ T-cell phenotypes	Flow cytometry	Melanoma and NSCLC	Increased density of cytolytic [73], PD-1+CD8+ T cells [77] and TCF7+ memory-like CD8+ T cells [78] were positively associated with ICB response.	
	Treg	Flow cytometry, IHC, or RNA seq	Multiple	Higher frequency of Treg was linked to unresponsiveness to [79] and hyperprogression after ICB [80].	No direct evidence in HCC yet.
	Macrophages	IHC, flow cytometry, or RNA seq	Multiple	TAM [81] and MDSC [79] is associated with unresponsiveness to PD-1 ICB.	No direct evidence in HCC yet.
Extra-tumoral	T-cell clonality	TCR repertoire sequencing	Multiple	Pretreatment TCR diversity and on-therapy TCR clonal expansion were correlated with clinical benefit [82,83,84].	No direct evidence in HCC yet.
	T-cell phenotypes	Flow cytometry, CyTOF	Melanoma	Higher T-cell reinvigoration [85] and T-cell activation [86] were associated with clinical outcome after anti-PD-1 therapy.	No direct evidence in HCC yet.
	MDSCs	Flow cytometry	Melanoma	Peripheral blood level of MDSCs correlated with poor anti-CTLA-4 response [87,88,89,90].	No direct evidence in HCC yet.
	Neutrophils/leukocytes	Flow cytometry	Multiple	Higher peripheral blood neutrophil/lymphocytes ratios were associated with decreased PFS and OS after ICB treatment [91,92,93,94].	No direct evidence in HCC yet.
	Treg	Flow cytometry	Melanoma	High baseline frequency [89], on-therapy increased [95] or decreased [96] in frequency of circulating Treg was associated with disease control upon ICB.	No direct evidence in HCC yet.
	LDH	Serum LDH detection	Melanoma and NSCLC	Baseline or on-therapy change of serum LDH levels correlated with OS of ICB-treated patients [89,96,97,98,99,100].	No direct evidence in HCC yet.
	Exosomal PD-L1	Exosome purification and characterization	Melanoma	Increased increase in circulating exosomal PD-L1 during early stages of treatment, as an indicator of the adaptive response of the tumour cells to T cell reinvigoration, stratifies clinical responders from non-respondersIncrease in circulating exosomal PD-L1 during early stages of treatment positively correlated with clinical response to anti-PD-1 therapy [47].	No direct evidence in HCC yet.
	cfDNA	cfDNA isolation followed by WES and WGS	Multiple	Specific mutations and TMB detected from circulating cfDNA associated with response to ICB [101,102].	Hypermutated circulating tumor DNA was associated with clinical outcome in 69 ICB-treated cancer patients (includes 3 HCC patients) [101].
	Gut microbiome	PCR or 16S rRNA gene sequencing	Multiple	Specific or diversity of gut microbiome was associated with response to ICB [103,104,105,106,107].	No direct evidence in HCC yet.

PD-L1, programmed cell death 1 ligand; PD-1, programmed cell death 1; HCC, hepatocellular carcinoma; seq, sequencing; ICB, immune checkpoint blockade that include anti-PD-1/PD-L1/CTLA-4 unless specified; TMB, tumor mutational burden; NGS, next-generation sequencing; WES, whole exome sequencing; WGS, whole genome sequencing; TME, tumor microenvironment; iCCA, intrahepatic cholangiocarcinoma; TILs, tumor-infiltrating lymphocytes; IHC, immunohistochemistry; TCR, T cell receptor; HLA, human leukocyte antigen; NSCLC, non-small cell lung cancer; CyTOF, Cytometry by Time-of-Flight; PFS, progression-free survival; OS, overall survival; MDSCs, myeloid derived suppressor cells; Treg, regulatory T cells; LDH, lactate dehydrogenase; cfDNA, cell-free DNA.

**Table 3 cancers-11-01926-t003:** Immune checkpoint combination therapy clinical trials in HCC.

Study Name	Phase	Target	Treatments	Estimated Enrolment ^	ORR (%)	DCR (%)	PFS (Median, mo)	OS (Median, mo)	Adverse Effect † > Grade 3
ICB-ICB Combination trials									
NCT01658878 (CheckMate040) * [140]	I/II	PD-1 + CTLA-4	Nivolumab + Ipilimumab	148	31%	49%	NR	40% (24-mo)	37%
NCT03298451 (HIMALAYA) [141]	III	PD-L1 + CTLA-4	Durvalumab versus Durvalumab + Tremelimumab vs. Sorafenib	1310	T.B.A.	T.B.A.	T.B.A.	T.B.A.	T.B.A.
NCT03680508	II	PD-1 + TIM3	TSR-042 + TSR-022	42	Not recruiting yet	T.B.A.	T.B.A.	T.B.A.	TBA
ICB-others Combination trials									
NCT02519348 [25]	I/II	PD-L1 alone or CTLA4 alone or PD-L1 + CTLA-4or PD-L1 + VEGF	Durvalumabor Tremelimumab or Durvalumab + Tremelimumab or Durvalumab + Bevacizumab	545	15% (6/40 patients)	57.5% #	NR	NR	20% (8/40 patients)
NCT03434379 (IMBrave150) [142]	III	PD-L1 + VEGF	Atezolizumab + Bevacizumab vs. Sorafenib	501 (336 vs. 165)	27% vs. 12%	74% vs. 55%	6.8 vs. 4.3	NE vs. 13.2	57% vs. 55%
NCT02715531 (Arm A) [22]	Ib	PD-L1 + VEGF	Atezolizumab+ Bevacizumab	104	36%	71%	7.3	17.1	27%
NCT02715531 (Arm F) [22]	Ib	PD-L1 + VEGF	Atezolizumab + Bevacizumab vs. Atezolizumab	60 59	20% 17%	67% 49%	5.6 3.4	NR	37% 14%
NCT03755791 (COSMIC-312) [143]	III	PD-L1+ mTKIs	Atezolizumab + Cabozantinib vs. Sorafenib vs. Cabozantinib	740	T.B.A.	T.B.A.	T.B.A.	T.B.A.	T.B.A.
NCT03006926 (Keynote 524) [144]	Ib	PD-1 + mTKIs	Pembrolizumab + Lenvatinib	30	36.7%	90%	9.7 (TTP)	14.6	73%
NCT03713593 (LEAP-002) [145]	III	PD-1 + mTKIs	Pembrolizumab + Lenvatinib vs. Lenvatinib	750	TBA	TBA	TBA	TBA	TBA
NCT03289533 [146]	I	PD-L1 + mTKIs	Avelumab + Axitinib	22	13.6%	68.2%	5.5	12.7	72.7%
NCT03092895 [147]	II	PD-1 + FOLFOX4 or GEMOX)	SHR-1210 + FOLFOX4 or GEMOX	34 (HCC patients)	26.5%	79.4%	5.5	NR	85.3%
NCT03071094	I/II	PD-1 + oncolytic virus	Nivolumab + Pexa-Vec	30	TBA	TBA	TBA	TBA	TBA

^, most updated from clinicaltrials.gov as of August 2019; *n,* number of patients; ORR, overall response rate; DCR, disease control rate; PFS, progression-free survival; OS, overall survival; mo, months; †, treatment-related adverse effects. *, Divided to three arms: Arm A: Nivolumab 1 mg/kg + Ipilimumab 3 mg/kg Q3W (4 doses); Arm B: Nivolumab 3 mg/kg + Ipilimumab 1 mg/kg Q3W (four doses), each followed by Nivolumab 240 mg Q2W, or Arm C: Nivolumab 3 mg/kg Q2W + Ipilimumab 1 mg/kg Q6W; NR, not reported; T.B.A., To be announced; #, CR+ PR + SD > 16 weeks; mTKIs, multitargeted tyrosine kinase inhibitors; NE, non-estimable; TTP, time to progression.

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
