# Peer review of "Rationale of Immunotherapy in Hepatocellular Carcinoma and Its Potential Biomarkers"

_cancers, 2019, doi:10.3390/cancers11121926_

Round 1

Reviewer 1 Report

Although no immunotherapy is not approved for HCC, immunotherapy in HCC is currently under investigation in many clinical trials. Therefore, this review paper seems very timely, and covers latest results and news release of clinical trials of immunotherapy for HCC.

2.2.1 the title “ICB abd ICB combination” should be changed to “anti-PD-1/PD-L1 plus anti-CTLA-4 combination” or something.

Page 15 2.5 “IRAE”

In this session, “IRAE” is used. It should be correct to “irAE”.

I possible, data on OS and/or PFS should be included in Table 1 and 2.

Author Response

Point by point response to reviewers' comments:

We would like to thank the reviewers for their expert opinions and excellent suggestions to improve our manuscript. We have since made changes which were underlined in the revised manuscript and detailed with the point-by-point response (R) to each comment below. In addition, following major announcements and reports from ESMO Asia 2019, we have added the result from IMBrave150 as well as a related mechanistic study by Shigeta et al Hepatology 2019. We have also included a recent report from our own work showing correlation between irAEs with response to ICB in 114 HCC patients. All of these additional important information, which we believe could tremendously improve the quality of this review, was underlined in the revised manuscript for your kind reference and review. In view of the comments on editing of English language and style, each of the co-authors has thoroughly checked the manuscript and performed another round of proof reading and editing. On a separate note, as one of the co-authors- Dr David Tai has been instrumental in the revision process, we have since readjusted the authorship position to make him the first author for this manuscript.

Reviewer #1:

Although no immunotherapy is not approved for HCC, immunotherapy in HCC is currently under investigation in many clinical trials. Therefore, this review paper seems very timely, and covers latest results and news release of clinical trials of immunotherapy for HCC.

R: We thank the reviewer for the kind comments.

2.2.1 the title “ICB abd ICB combination” should be changed to “anti-PD-1/PD-L1 plus anti-CTLA-4 combination” or something.

R: The reason why we prefer to use ICB and ICB combination is because first we have already defined ICB and acknowledged that majority of trials in HCC involve anti-PD-1/PD-L1 and anti-CTLA-4 in section 2.1: Immune checkpoint blockade (ICB) therapy. Also even though these ICBs are the most commonly tested in HCC at the moment, it does not limit the combination with other ICBs such as with anti-Tim-3 (One on-going trial with anti-PD-1+anti-Tim3 was listed in Table 3). For this purpose, we feel it is still better to keep the scope wider and stick to “ICB and ICB combination”, rather than defining which ICBs.

Page 15 2.5 “IRAE”. In this session, “IRAE” is used. It should be correct to “irAE”.

R: Noted on the discrepancy, we have since checked through the entire manuscript to make sure we use “irAEs” consistently. All changes were marked with underlined.

If possible, data on OS and/or PFS should be included in Table 1 and 2.

R: This is a good suggestion, we have since included the median OS and PFS in Table 1 and Table 3 with all the available data. New data added was marked with underlined.

Reviewer 2 Report

Manuscript entitled, "Rationale of Immunotherapy in Hepatocellular Carcinoma and its Potential Biomarkers" is a very exhaustive compilation of clinical research reports using standalone as well as combination immunotherapy with immune checkpoint blockers for the treatment of Hepatocellular carcinoma. Though the manuscript is well written and neatly structured, yet it didn’t provide much novel insight/idea [as similar topic also have been published in two publications -- Therapeutic Advances in Medical Oncology 11 (2019): 1758835919862692 & Cancers 11.8 (2019): 1078]. In addition, there are many minor and major points/errors that need careful revision before the manuscript can be considered for publication. Below are the comments:

The authors did not describe PD1/PD- L1 status in the treated patients. How about the level of related targets level in the treated patients? Can the authors introduce some of condition of the patients who need to receive the immunotherapeutic strategies? How about the effort of tumor mutational burden and neoantigen toward to the HCC treatment? Could the authors also provide some discussion about this topic? The sequence of the statement is a little hard for me to understand. I believe the authors need to describe the background first and the turn to illustrate the therapy strategy. Thus, I suggest that the whole 2.4 section need to move forward and those similar parts (e.g. the PD-1/PD-L1) in previous section need to rearrange. For the drug information, different drug with different effects, could you provide more information about those drugs and then simply analyze that because it can help us to better understand. (e.g. Nivolumab; Sorafenib; pembrolizumab and so on ) Line 38-40: Are these irAEs the result of ICBs treatments? if so, then specify this in the statement. Line 46: In the heading, "Current landscape of immunotherapy therapy in HCC", I think there is no need of the word "therapy". In Table 1, some of the treatments (not all) have been shown to be conducted in comparison to either another drug or placebo. It will be more informative if the standard is shown for all the treatments. Table 1: it looks like study 15 is a part of study 11 with an emphasis on Sorafenib-experienced Asian cohort analysis. This information is missing in the table. it is important to mention this piece of information because both the studies have same study name and it is confusing for the readers. Line 59: T.B.A is not required in foot-note as there is no such information in table1. Line 90: “Despite both studies not meeting its primary endpoints," should be, “Despite both studies not meeting their primary endpoints," Line 105, 115: Instead of writing as "will be discussed below", it is better to mention the exact section number, where that particular information ‘will be discussed’ in detail. Line 251-252: "Combination ACT with checkpoint inhibitor has yet to be explored in HCC." should be, "Combination ACT with checkpoint inhibitor is yet to be explored in HCC." Line 28 A variety of strategies have been explored; cytokine -- The ; should be changed to be a : Line 151-161 The size and type of font have problems The size and type of font (reference 188-193 in the text) have problems. In 2.1.1 of this manuscript, the authors mentioned ‘disease control rate (DCR)’, this data could be added into table 1 to make the content more comprehensive. In addition, the authors should also pay more attentions on the format; different font and size are found in the last paragraph of 2.5 and the first paragraph of 2.1.1. The description of ‘Clinical relevance’ and ‘Relevance to HCC’ columns in table 3 are a little bit verbose. Authors should also consider some recent publications in this field to further enrich the contents of the manuscript as a review article -- Current Protein and Peptide Science 20: 82-91; International Journal of Molecular Sciences 18(12): 2540; ADMET and DMPK 5(3): 159-172; Cancers 10(3): 82

Author Response

Point by point response to reviewers' comments:

We would like to thank the reviewers for their expert opinions and excellent suggestions to improve our manuscript. We have since made changes which were underlined in the revised manuscript and detailed with the point-by-point response (R) to each comment below. In addition, following major announcements and reports from ESMO Asia 2019, we have added the result from IMBrave150 as well as a related mechanistic study by Shigeta et al Hepatology 2019. We have also included a recent report from our own work showing correlation between irAEs with response to ICB in 114 HCC patients. All of these additional important information, which we believe could tremendously improve the quality of this review, was underlined in the revised manuscript for your kind reference and review. In view of the comments on editing of English language and style, each of the co-authors has thoroughly checked the manuscript and performed another round of proof reading and editing. On a separate note, as one of the co-authors- Dr David Tai has been instrumental in the revision process, we have since readjusted the authorship position to make him the first author for this manuscript.

Reviewer #2:

Manuscript entitled, "Rationale of Immunotherapy in Hepatocellular Carcinoma and its Potential Biomarkers" is a very exhaustive compilation of clinical research reports using standalone as well as combination immunotherapy with immune checkpoint blockers for the treatment of Hepatocellular carcinoma. Though the manuscript is well written and neatly structured, yet it didn’t provide much novel insight/idea [as similar topic also have been published in two publications -- Therapeutic Advances in Medical Oncology 11 (2019): 1758835919862692 & Cancers 11.8 (2019): 1078]. In addition, there are many minor and major points/errors that need careful revision before the manuscript can be considered for publication.

R: We appreciate the assessment by the reviewer and would aim to improve the quality of our manuscript and address the concerns as detailed below. First of all, we would like to comment on the novelty of our review article. Indeed, many of the review articles including the two that were mentioned above provide a general description of the current HCC immunotherapy trials and landscapes that look highly similar across. Our review article however went a step further to comprehensively describe and summarize the potential biomarkers (Table 2) with the potential implications for HCC. Besides, we also explain the mechanism of actions of immunotherapy based on these biomarkers. In addition, for each combination, we also explain their mechanisms of actions based on the cancer immunity cycle illustrated in Figure 1. With this comprehensive and detailed review on the immunotherapy in HCC, we do believe that our article provides novelty and useful information for the community.

Next, we would also like to take this opportunity to specially thank the reviewer for his/her time and effort to provide such thorough review of our article and spotted many weaknesses and errors, which we could improve on. We were very impressed by the quality of review and the attentiveness to details by this reviewer. We apologize for overlooking these errors and have since corrected them one by one as detailed below. All changes were marked with underlined.

Below are the comments:

The authors did not describe PD1/PD- L1 status in the treated patients. How about the level of related targets level in the treated patients?

R: Indeed, we only mentioned that PD-L1 status is not predictive of response to ICB for HCC patients. We now included a sentence under section 2.2.1 to summarize on previous reports for expression of PD-L1/PD-1 in HCC.

“In HCC, PD-L1 was found to be expressed by the tumour cells [38] and macrophages [39], both of which were associated with poor disease prognosis; while PD-1 was expressed mainly by the T cells including regulatory T cells (Treg) [40,41]. It has also been shown that the PD-L1 expression in HCC is generally low (~10% by tumour cells) and highly heterogeneous across different anti-PD-L1 staining antibodies used [42]. Indeed, tumour PD-L1 expression was not associated with response to anti-PD-1 therapy in both CheckMate040 and Keynote224 trials in HCC [11,12].”

Can the authors introduce some of condition of the patients who need to receive the immunotherapeutic strategies?

R: This is a very good suggestion, in fact there are currently lack of strong biomarker to inform the clinical decision of which HCC patient should receive and would likely respond to immunotherapy. We have now provided our views in section 3: Future perspectives as underlined below:

“Based on the biomarkers studies with limited available data in HCC (Table 2), we would conclude that phenotypes of tumour infiltrating T cells particularly the cytotoxic CD8 T cells, appear to be an important determining factor for response to immunotherapy. As HCC tumours are enriched with Treg [40] and generally not well infiltrated by immune cells [56,115], strategies to inhibit Treg and enhance T cells infiltration, such as that with the anti-angiogenesis agents, would be expected to be synergetic to ICB therapy. Given the recent success of Phase III trial in HCC using ICB plus anti-angiogenesis agent (IMbrave150) [143], it is increasingly clear that a combination strategy with clear scientific rationale would be necessary for enhanced clinical response for immunotherapy.”

How about the effort of tumor mutational burden and neoantigen toward to the HCC treatment? Could the authors also provide some discussion about this topic?

R: This is another important point. In fact, we have described this under section 2.2.2. Tumour mutational burden (TMB) and specific genomic mutations, whereby we described: TMB level is considered moderate for HCC with a good correlation between TMB and PD-L1 tumour expression levels, suggestive of possibly a moderate response to immunotherapy [52].

There is currently no available data on neoantigen hence this was not mentioned.

The sequence of the statement is a little hard for me to understand. I believe the authors need to describe the background first and the turn to illustrate the therapy strategy. Thus, I suggest that the whole 2.4 section need to move forward and those similar parts (e.g. the PD-1/PD-L1) in previous section need to rearrange.

R: We think this is a very good suggestion and it makes the review article flow better. We have since moved the section on biomarkers and irAES up (now they are 2.2 and 2.3 respectively) and the review goes with the flow as below:

Introduction Current landscape of immunotherapy in HCC

2.1. Immune checkpoint blockade (ICB) therapy

2.2. Current knowledge on biomarkers for ICBs and its relevance in HCC

2.3. irAEs and its association with outcomes of ICBs in HCC

2.4. Current landscape and rationale of combination immunotherapy in HCC

2.5. Other immunotherapies and their potential as combination in HCC

Future Perspectives Concluding remarks

For the drug information, different drug with different effects, could you provide more information about those drugs and then simply analyze that because it can help us to better understand. (e.g. Nivolumab; Sorafenib; pembrolizumab and so on )

R: In fact, we did describe the mechanisms of actions of each therapy at the beginning of the section. To further improve on this, we included more description as underlined in each section when describing the class of drugs.

Line 38-40: Are these irAEs the result of ICBs treatments? if so, then specify this in the statement.

R: Yes, they are. We have revised it as “Moreover, 15-25% of these ICBs-treated patients experienced grade 3/4 treatment or immune-related adverse events (TRAEs or irAEs)…”

Line 46: In the heading, "Current landscape of immunotherapy therapy in HCC", I think there is no need of the word "therapy".

R: Indeed, we have deleted the word “therapy”.

In Table 1, some of the treatments (not all) have been shown to be conducted in comparison to either another drug or placebo. It will be more informative if the standard is shown for all the treatments.

R: We agree this will be informative. We have now included this information in Table 1.

Table 1: it looks like study 15 is a part of study 11 with an emphasis on Sorafenib-experienced Asian cohort analysis. This information is missing in the table. it is important to mention this piece of information because both the studies have same study name and it is confusing for the readers.

R: In fact, we did mention this under the treatments column but yes we agree it looks like the same study and can be confusing. We have now moved the (Asian cohort analysis) to the first column.

Line 59: T.B.A is not required in foot-note as there is no such information in table1.

R: Yes, thank you for spotting this error, we have deleted this.

Line 90: “Despite both studies not meeting its primary endpoints," should be, “Despite both studies not meeting their primary endpoints,"

R: Yes, we have changed “its” to “their”.

Line 105, 115: Instead of writing as "will be discussed below", it is better to mention the exact section number, where that particular information ‘will be discussed’ in detail.

R: Indeed, since we have already mentioned this in the beginning of section 2.1, we have changed the sentence in the first paragraph to “Combination strategies utilizing ICBs will be described in greater detail below in section 2.4.” We do not repeat this sentence again under each ICB types.

Line 251-252: "Combination ACT with checkpoint inhibitor has yet to be explored in HCC." should be, "Combination ACT with checkpoint inhibitor is yet to be explored in HCC."

R: We have changed “has” to “is”.

Line 28 A variety of strategies have been explored; cytokine -- The ; should be changed to be a :

R: We have changed “;” to “:”.

Line 151-161 The size and type of font have problems The size and type of font (reference 188-193 in the text) have problems.

R: These have been corrected.

In 2.1.1 of this manuscript, the authors mentioned ‘disease control rate (DCR)’, this data could be added into table 1 to make the content more comprehensive.

R: We agree this will be informative. We have now included this information in Table 1.

In addition, the authors should also pay more attentions on the format; different font and size are found in the last paragraph of 2.5 and the first paragraph of 2.1.1.

R: We have checked and corrected the font size and format though out the manuscript.

The description of ‘Clinical relevance’ and ‘Relevance to HCC’ columns in table 3 are a little bit verbose.

R: As the biomarkers can be complex and some essential information might be helpful, we have tried to summarize this table without compromising on the important information.

Authors should also consider some recent publications in this field to further enrich the contents of the manuscript as a review article -- Current Protein and Peptide Science 20: 82-91; International Journal of Molecular Sciences 18(12): 2540; ADMET and DMPK 5(3): 159-172; Cancers 10(3): 82

R: Indeed, we have now incorporated several key areas especially on post-translational modifications and transport of proteins in the regulation of PD-L1 expression in tumour (underline in section 2.2.1. PD-L1 expression). We however find the last paper suggested: Cancers, 10(3): 82 (on the roles of protein tyrosine phosphatases in Hepatocellular Carcinoma) has little relevance to immune microenvironment or to immunotherapy and hence has excluded it from the current scope.

Round 2

Reviewer 2 Report

I have no further comment on the revised manuscript.